# CONDITIONAL DIFFUSION DISTILLATION

## ABSTRACT

Generative diffusion models provide strong priors for text-to-image generation and thereby serve as a foundation for conditional generation tasks such as image editing, restoration, and super-resolution. However, one major limitation of diffusion models is their slow sampling time. To address this challenge, we present a novel conditional distillation method designed to supplement the diffusion priors with the help of image conditions, allowing for conditional sampling with very few steps. We directly distill the unconditional pre-training in a single stage through joint-learning, largely simplifying the previous two-stage procedures that involve both distillation and conditional finetuning separately. Furthermore, our method enables a new parameter-efficient distillation mechanism that distills each task with only a small number of additional parameters combined with the shared frozen unconditional backbone. Experiments across multiple tasks including super-resolution, image editing, and depth-to-image generation demonstrate that our method outperforms existing distillation techniques for the same sampling time. Notably, our method is the first distillation strategy that can match the performance of the much slower fine-tuned conditional diffusion models.

## 1 INTRODUCTION

Text-to-image diffusion models (Saharia et al., 2022b; Rombach et al., 2022; Ramesh et al., 2022) trained on large-scale data (Lin et al., 2014; Schuhmann et al., 2022) have significantly dominated generative tasks by delivering impressive high-quality and diverse results. A newly emerging trend is to use the diffusion prior of pre-trained text-to-image generative models to guide the generated results with external image conditions for traditional image-to-image transformation tasks such as image manipulation, enhancement, or super-resolution (Meng et al., 2021; Zhang & Agrawala, 2023). Among these transformation processes, the diffusion prior introduced by pre-trained models is shown to be capable of greatly promoting the visual quality of the conditional image generation results (Brooks et al., 2023).

However, diffusion models heavily rely on an iterative refinement process (Song et al., 2020c; Saharia et al., 2022c;a; Whang et al., 2022; Delbracio & Milanfar, 2023) that often demands a substantial number of iterations, which can be challenging to accomplish efficiently. Their reliance on the number of iterations further increases for high-resolution image synthesis. For instance, in state-of-the-art text-to-image latent diffusion models (Rombach et al., 2022), achieving optimal visual quality typically requires $20 - 200$ sampling steps (function evaluations), even with advanced sampling methods (Lu et al., 2022a; Karras et al., 2022). The slow sampling time significantly impedes practical applications of the aforementioned conditional diffusion models.

Recent efforts to accelerate diffusion sampling predominantly employ distillation methods (Luhman & Luhman, 2021; Salimans & Ho, 2022; Song et al., 2023). These methods achieve significantly faster sampling, completing the process in just $4 - 8$ steps, with only a marginal decrease in generative performance. Very recent works (Meng et al., 2023; Li et al., 2023) show that these strategies are even applicable for distilling pre-trained large-scale text-to-image diffusion models. Based on these distillation techniques, a two-stage distillation procedure (Meng et al., 2023) can be used for distilling conditional diffusion models —either distillation-first or conditional finetuning-first. These two procedures offer different advantages in terms of cross-task flexibility and learning difficulty, but their generated results (Meng et al., 2023) are generally better than those of the undistilled conditional diffusion model when given the same sampling time.

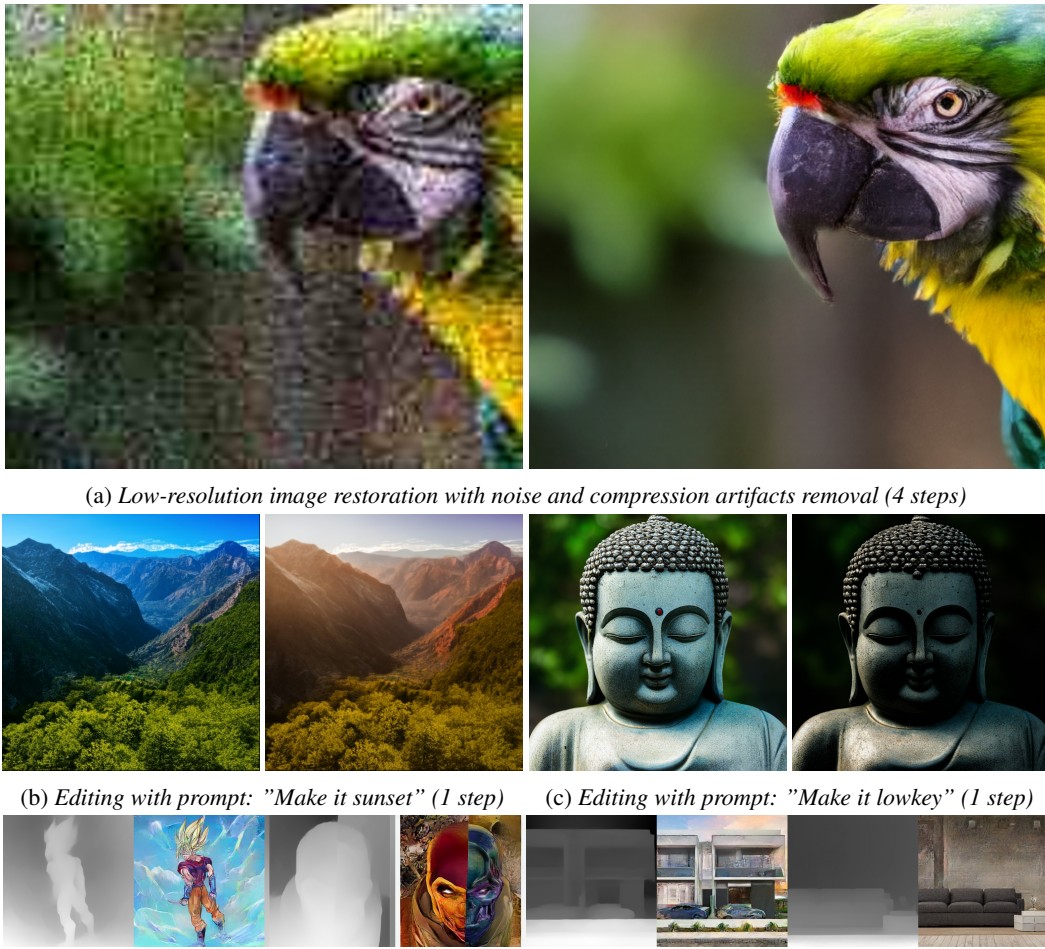

(a) *Low-resolution image restoration with noise and compression artifacts removal (4 steps)*

(b) *Editing with prompt: "Make it sunset" (1 step)* (c) *Editing with prompt: "Make it lowkey" (1 step)*

(d) *Generating images from the depth map. (4 steps)*

Figure 1: Our method distills a conditional diffusion model directly from the unconditional model. We show the generated results of our distilled model in various conditional tasks, which demonstrates the capability of our proposed method in replicating diffusion priors in a short sampling time.

In this paper, we introduce a new distillation approach for distilling a conditional diffusion model from a pre-trained unconditional diffusion one. Unlike the previous two-stage distillation procedure, our method only has a single stage that starts from the unconditional pretraining and ends with the distilled conditional diffusion model. In Figure 1, we show that our distilled model can predict high-quality results in $1-4$ sampling steps by using the hints from the given image conditions. This simplified learning eliminates the need for the original text-to-image data, a requirement in previous distillation procedures (*i.e.*, those that first distill the unconditional text-to-image model) thereby making our method more practical. Additionally, our formulation avoids sacrificing the diffusion prior in the pre-trained model, that commonly occurs in the first stage of the finetuning-first procedure. Extensive experimental results show that our distilled model outperforms previous distillation methods in both visual quality and quantitative performance, when given the same sampling time.

Parameter-efficient distillation methods for conditional generation are a relatively understudied area. We demonstrate that our methodology enables a new parameter-efficient distillation mechanism. It can transform and accelerate an unconditional diffusion model for conditional tasks by incorporating a limited number of additional learnable parameters. In particular, our formulation allows the integration with various existing parameter-efficient tuning algorithms, including T2I-Adapter (Mou et al., 2023) and ControlNet (Zhang & Agrawala, 2023). Our distillation process learns to replicate diffusion priors for conditional tasks with few iterative refinements, using both the newly added learnable parameters of the conditional adapter and the frozen parameters of the original diffusion model. This new paradigm significantly improves the practicality of different conditional tasks.

## 2 BACKGROUND

**Continuous-time VP diffusion model.** A continuous-time variance-preserving (VP) diffusion model (Sohl-Dickstein et al., 2015; Ho et al., 2020) is a special case of diffusion models[1]. It has latent variables $\{\mathbf{z}_t | t \in [0, T]\}$ specified by a noise schedule comprising differentiable functions $\{\alpha_t, \sigma_t\}$ with $\sigma_t^2 = 1 - \alpha_t^2$. The clean data $\mathbf{x} \sim p_{\text{data}}$ is progressively perturbed in a (forward) Gaussian process as in the following Markovian structure:

$$q(\mathbf{z}_t|\mathbf{x}) = \mathcal{N}(\mathbf{z}_t; \alpha_t \mathbf{x}, \sigma_t^2 \mathbf{I}), \text{ and } q(\mathbf{z}_t|\mathbf{z}_s) = \mathcal{N}(\mathbf{z}_t; \alpha_{t|s} \mathbf{z}_s, \sigma_{t|s}^2 \mathbf{I}), \tag{1}$$

where $0 \le s < t \le 1$ and $\alpha_{t|s}^2 = \alpha_t / \alpha_s$. Here the latent $\mathbf{z}_t$ is sampled from the combination of the clean data and random noise by using the reparameterization trick (Kingma & Welling, 2013), which has $\mathbf{z}_t = \alpha_t \mathbf{x} + \sigma_t \epsilon$.

**Deterministic sampling.** The aforementioned diffusion process that starts from $\mathbf{z}_0 \sim p_{\text{data}}(\mathbf{x})$ and ends at $\mathbf{z}_T \sim \mathcal{N}(0, \mathbf{I})$ can be modeled as the solution of an stochastic differential equation (SDE) (Song et al., 2020c). The SDE is formed by a vector-value function $f(\cdot, \cdot) : \mathbb{R}^d \to \mathbb{R}^d$, a scalar function $g(\cdot) : \mathbb{R} \to \mathbb{R}$, and the standard Wiener process $\mathbf{w}$ as:

$$d\mathbf{z}_t = f(\mathbf{z}_t, t)dt + g(t)d\mathbf{w}. \tag{2}$$

The overall idea is that the reverse-time SDE that runs backwards in time, can generate samples of $p_{\text{data}}$ from the prior distribution $\mathcal{N}(0, \mathbf{I})$. This reverse SDE is given by

$$d\mathbf{z}_t = [f(\mathbf{z}_t, t) - g(t)^2 \nabla_{\mathbf{z}} \log p_t(\mathbf{z}_t)]dt + g(t)d\bar{\mathbf{w}}, \tag{3}$$

where the $\bar{\mathbf{w}}$ is a also standard Wiener process in reversed time, and $\nabla_{\mathbf{z}} \log p_t(\mathbf{z}_t)$ is the score of the marginal distribution at time $t$. The score function can be estimated by training a score-based model $s_\theta(\mathbf{z}_t, t) \approx \nabla_z \log p_t(\mathbf{z}_t)$ with score-matching (Song et al., 2020b) or a denoising network $\hat{\mathbf{x}}_\theta(\mathbf{z}_t, t)$ (Ho et al., 2020):

$$s_\theta(\mathbf{z}_t, t) := (\alpha_t \hat{\mathbf{x}}_\theta(\mathbf{z}_t, t) - \mathbf{z}_t) / \sigma_t^2. \tag{4}$$

Such backward SDE satisfies a special ordinary differential equation (ODE) that allows deterministic sampling given $\mathbf{z}_T \sim \mathcal{N}(0, \mathbf{I})$. This is known as the *probability flow* ODE (Song et al., 2020c) and is given by

$$d\mathbf{z}_t = [f(\mathbf{z}_t, t) - \frac{1}{2} g^2(t) s_\theta(\mathbf{z}_t, t)]dt, \tag{5}$$

where $f(\mathbf{z}_t, t) = \frac{d \log \alpha_t}{dt} \mathbf{z}_t$, $g^2(t) = \frac{d\sigma_t^2}{dt} - 2 \frac{d \log \alpha_t}{dt} \sigma_t^2$ with respect to $\{\alpha_t, \sigma_t\}$ and $t$ according to Kingma et al. (2021). This ODE can be solved numerically with diffusion samplers like DDIM (Song et al., 2020a), where starting from $\hat{\mathbf{z}}_T \sim \mathcal{N}(0, \mathbf{I})$, we update for $s = t - \Delta t$:

$$\hat{\mathbf{z}}_s := \alpha_s \hat{\mathbf{x}}_\theta(\hat{\mathbf{z}}_t, t) + \sigma_s (\hat{\mathbf{z}}_t - \alpha_t \hat{\mathbf{x}}_\theta(\hat{\mathbf{z}}_t, t)) / \sigma_t, \tag{6}$$

till we reach $\hat{\mathbf{z}}_0$.

**Diffusion models parametrizations.** Leaving aside the aforementioned way of parametrizing diffusion models with a denoising network (signal prediction) or a score model (noise prediction equation 4), in this work, we adopt a parameterization that mixes both the score (or noise) and the signal prediction. Existing methods include either predicting the noise $\hat{\epsilon}_\theta(\mathbf{x}_t, t)$ and the signal $\hat{\mathbf{x}}_\theta(\mathbf{z}_t, t)$ separately using a single network (Dhariwal & Nichol, 2021), or predicting a combination of noise and signal by expressing them in a new term, like the velocity model $\hat{\mathbf{v}}_\theta(\mathbf{z}_t, t) \approx \alpha_t \epsilon - \sigma_t \mathbf{x}$ (Salimans & Ho, 2022). Note that one can derive an estimation of the signal and the noise from the velocity one,

$$\hat{\mathbf{x}} = \alpha_t \mathbf{z}_t - \sigma_t \hat{\mathbf{v}}_\theta(\mathbf{z}_t, t), \text{ and } \hat{\epsilon} = \alpha_t \hat{\mathbf{v}}_\theta(\mathbf{z}_t, t) + \sigma_t \mathbf{z}_t. \tag{7}$$

Similarly, DDIM update rule (equation 6) can be rewritten in terms of the velocity parametrization:

$$\hat{\mathbf{z}}_s := \alpha_s (\alpha_t \hat{\mathbf{z}}_t - \sigma_t \hat{\mathbf{v}}_\theta(\hat{\mathbf{z}}_t, t)) + \sigma_s (\alpha_t \hat{\mathbf{v}}_\theta(\hat{\mathbf{z}}_t, t) + \sigma_t \hat{\mathbf{z}}_t). \tag{8}$$

---

[1]What we discussed based on the variance preserving (VP) form of SDE (Song et al., 2020c) is equivalent to most general diffusion models like Denoising Diffusion Probabilistic Models (DDPM) (Ho et al., 2020).

**Self-consistency property.** To accelerate inference, Song et al. (2023) introduced the idea of consistency models. Let $s_\theta(\cdot, t)$ be a pre-trained diffusion model trained on data $\mathbf{x} \sim \mathcal{O}_{data}$. Then, a consistency function $f_\phi(\mathbf{z}_t, t)$ should satisfy that (Song et al., 2023),

$$f_\phi(\mathbf{z}_t, t) = f_\phi(\mathbf{z}_{t'}, t'), \ \forall t, t' \in [0, T], \text{and } f_\phi(\mathbf{x}, 0) = \mathbf{x}, \tag{9}$$

where $\{\mathbf{z}_t\}_{t \in [0,T]}$ is the solution trajectory of the PF ODE (equation 5). The consistency function can be distilled from the pretrained model by enforcing the above self-consistency property. In practice, $f_\phi(\mathbf{z}_t, t)$ is usually a denoising network that is distilled from a pre-trained diffusion model.

## 3 RELATED WORK

To reduce the sampling time of diffusion models, Luhman & Luhman (2021) proposed to learn a single-step student model from the output of the original (teacher) model using multiple sampling steps. However, this method requires to run the full inference with many sampling steps during training which make it poorly scalable. Inspired by this, Progressive Distillation (Salimans & Ho, 2022) and its variants, including Guided Distillation (Meng et al., 2023) and SnapFusion (Li et al., 2023), use a progressive learning scheme for improving the learning efficiency. A student model learns to predict in one step the output of two steps of the teacher model. Then, the teacher model is replaced by the student model, and the procedure is repeated to progressively distill the mode by halving the number of required steps. We demonstrate our method by comparing with a fine-tuned version of Guided Distillation (Meng et al., 2023) on the conditional generation tasks. We also note that the learning strategies like classifier free guidance aware distillation, used by Meng et al. (2023) and Li et al. (2023), is orthogonal to our method, and they could be still applicable in our framework.

Song et al. (2023) introduced Consistency Models, a single-step generative approach that learns from a pre-trained diffusion model. The learning is achieved by enforcing a self-consistency in the predicted signal space. However, learning consistency models for conditional generation has yet to be thoroughly studied. In this paper, we compare our method against consistency models in a *brute force* way that learns to enforce the self-consistency in a fine-tuned conditional diffusion model. We will later show that this is less effective than our conditional distillation.

## 4 METHOD

### 4.1 FROM AN UNCONDITIONAL TO A CONDITIONAL ARCHITECTURE

In order to utilize the image generation prior encapsulated by the pre-trained unconditional[2] diffusion model, we first propose to adapt the unconditional diffusion model into a conditional version for the conditional data $(\mathbf{x}, c) \sim p_{\text{data}}$. Similar to the zero initialization technique used by controllable generation (Nichol & Dhariwal, 2021; Zhang & Agrawala, 2023), our method adapts the unconditional pre-trained architecture by using an additional conditional encoder.

To elaborate, we take the widely used U-Net as the diffusion network. Let us introduce the conditional-module by duplicating the encoder layers of the pretrained network. Then, let $\boldsymbol{h}_\theta(\cdot)$ be the encoder features of the pretrained network, and $\boldsymbol{h}_\eta(\cdot)$ be the features on the additional conditional encoder. We define the new encoder features of the adapted model by

$$\boldsymbol{h}_\theta(\mathbf{z}_t)' = (1 - \mu)\boldsymbol{h}_\theta(\mathbf{z}_t) + \mu\boldsymbol{h}_\eta(c), \tag{10}$$

where $\mu$ is a learnable scalar parameter, initialized to $\mu = 0$. Starting from this zero initialization, we can adapt the unconditional architecture into a conditional one. Thus, our conditional diffusion model $\hat{\mathbf{w}}_\theta(\mathbf{z}_t, c, t)$ is the result of adapting the pre-trained unconditional diffusion model $\hat{\mathbf{v}}_\theta(\mathbf{z}_t, t)$ with the conditional features $\boldsymbol{h}_\eta(c)$.

### 4.2 CONDITIONAL DIFFUSION DISTILLATION

Our core idea is to optimize the adapted conditional diffusion model $\hat{\mathbf{w}}_\theta(\mathbf{z}_t, c, t)$ from $\hat{\mathbf{v}}_\theta(\mathbf{z}_t, t)$, so (i) it satisfies the self-consistency property in equation 9, and (ii) it jointly learns to generate samples from the conditional data. To motivate our approach, let us introduce the following general remark.

---

[2]The discussed unconditional models include text-conditioned image generation models, *e.g.*, StableDiffusion (Rombach et al., 2022) and Imagen (Saharia et al., 2022b), which are only conditioned on text prompts.

---

**Algorithm 1** Conditional Diffusion Distillation

---

**Input:** conditional data $(\mathbf{x}, c) \sim p_{\text{data}}$, adapted diffusion model $\hat{\mathbf{w}}_\theta(\mathbf{z}_t, c, t)$ with parameters $\theta$, learning rate $\eta$, distance functions $d_\epsilon(\cdot, \cdot)$ and $d_{\mathbf{x}}(\cdot, \cdot)$, and exponential moving average $\gamma$
$\boldsymbol{\theta}^- \leftarrow \boldsymbol{\theta}$ $\quad\quad\triangleright$*target network initlization*
**repeat**
$\quad$ Sample $(\mathbf{x}, c) \sim p_{\text{data}}$ and $t \sim [\Delta t, T]$ $\quad\quad\triangleright$*empirically $\Delta t = 1$*
$\quad$ Sample $\epsilon \sim \mathcal{N}(0, \mathbf{I})$
$\quad s \leftarrow t - \Delta t$
$\quad \mathbf{z}_t \leftarrow \alpha_t \mathbf{x} + \sigma_t \epsilon$
$\quad \hat{\mathbf{x}}_t \leftarrow \alpha_t \mathbf{z}_t - \sigma_t \hat{\mathbf{w}}_\theta(\mathbf{z}_t, c, t)$ $\quad\quad\triangleright$*signal prediction in equation 7*
$\quad \hat{\epsilon}_t \leftarrow \alpha_t \hat{\mathbf{w}}_\theta(\mathbf{z}_t, c, t) + \sigma_t \mathbf{z}_t$ $\quad\quad\triangleright$*noise prediction in equation 7*
$\quad \hat{\mathbf{z}}_s \leftarrow \alpha_s \hat{\mathbf{x}}_t + \sigma_s \hat{\epsilon}_t$ $\quad\quad\triangleright$*update rule in equation 8*
$\quad \hat{\epsilon}_s \leftarrow \alpha_s \mathbf{w}_{\theta^-}(\hat{\mathbf{z}}_s, c, t) + \sigma_s \hat{\mathbf{z}}_s$ $\quad\quad\triangleright$*noise prediction in equation 7*
$\quad \mathcal{L}(\theta, \theta^-) \leftarrow d_\epsilon(\hat{\epsilon}_t, \hat{\epsilon}_s) + d_{\mathbf{x}}(\mathbf{x}, \hat{\mathbf{x}}_t)$ $\quad\quad\triangleright$*the distillation loss in equation 11*
$\quad \boldsymbol{\theta} \leftarrow \boldsymbol{\theta} - \eta \nabla_{\boldsymbol{\theta}} \mathcal{L}(\boldsymbol{\theta}, \boldsymbol{\theta}^-)$
$\quad \boldsymbol{\theta}^- \leftarrow \text{stopgrad}(\gamma \boldsymbol{\theta}^- + (1 - \gamma)\boldsymbol{\theta})$ $\quad\quad\triangleright$*exponential moving average*
**until** convergence

---

**Remark 1.** *If a diffusion model, parameterized by $\hat{\mathbf{v}}_\theta(\mathbf{z}_t, t)$, satisfies the self-consistency property on the noise prediction $\hat{\epsilon}_\theta(\mathbf{z}_t, t) = \alpha_t \hat{\mathbf{v}}_\theta(\mathbf{z}_t, t) + \sigma_t \mathbf{z}_t$, then it also satisfies the self-consistency property on the signal prediction $\hat{\mathbf{x}}_\theta(\mathbf{z}_t, t) = \alpha_t \mathbf{z}_t - \sigma_t \hat{\mathbf{v}}_\theta(\mathbf{z}_t, t)$.*

The proof is a direct consequence of change of variables from noise into signal and is given in Appendix **??**. Based on this general remark, we claim that we can optimize the conditional diffusion model $\hat{\mathbf{w}}_\theta(\mathbf{z}_t, c, t)$ to jointly learn to enforce the self-consistency property on the noise prediction $\hat{\epsilon}_\theta(\mathbf{z}_t, c, t)$ and the new conditional generation $(\mathbf{x}, c) \sim p_{\text{data}}$ with the signal prediction $\hat{\mathbf{x}}_\theta(\mathbf{z}_t, c, t)$.

To elaborate the distillation learning, we denote the latent variables $\mathbf{z}_t$ as the randomly sampled latent variable (equation 1), and $\hat{\mathbf{z}}_s$ is the predicted latent variable that belongs to the same trajectory of $\mathbf{z}_t$ in the PF ODE (equation 5), integrates the adapted conditional diffusion model $\hat{\mathbf{w}}_\theta(\mathbf{z}_t, c, t)$. Inspired by Remark 1, we introduce the following training scheme.

**Training scheme.** Inspired by consistency models (Song et al., 2023), we use the exponential moving averaged parameters $\theta^-$ as the target network for stabilize training. Then, we seek to minimize the following training loss for conditional distillation:

$$\mathcal{L}(\theta) := \mathbb{E}[d_\epsilon(\underbrace{\hat{\epsilon}_{\theta^-}}_{\text{\# target network}}(\underbrace{\hat{\mathbf{z}}_s}_{\text{\# sampled using the empirical PF ODE}}, s, c), \underbrace{\hat{\epsilon}_\theta(\mathbf{z}_t, t, c)}_{\text{\# online network}})) + d_{\mathbf{x}}(\mathbf{x}, \hat{\mathbf{x}}_\theta(\mathbf{z}_t, t, c)], \quad (11)$$

where $d_\epsilon(\cdot, \cdot)$ and $d_{\mathbf{x}}(\cdot, \cdot)$ are two distance functions to measure difference in the noise space and in the signal space respectively. Note that the total loss is a balance between the conditional guidance given by $d_{\mathbf{x}}$, and the noise self-consistency property given by $d_\epsilon$.

The overall conditional distillation algorithm is presented in Algorithm 1, and illustrated in Figure 2. In the following, we will detail how we sample $\hat{\mathbf{z}}_s$ and discuss other relevant hyperparameters in our method (e.g., $d_{\mathbf{x}}$).

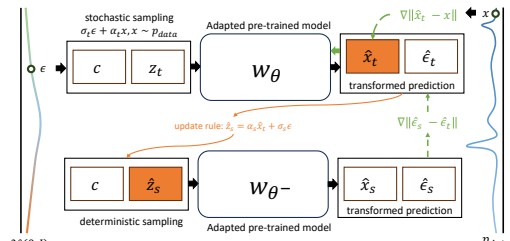

Figure 2: The diagram illustrates the distillation process of our proposed method. The green arrow denotes the gradient of the back propagation during learning.

**Prediction of $\hat{\mathbf{z}}_s$.** In the distillation process given by equation 11, the latent variable $\hat{\mathbf{z}}_s$ is achieved by running one step of a numerical ODE solver. Consistency models (Song et al., 2023) solve the ODE using the Euler solver, while progressive distillation (Salimans & Ho, 2022) and guided distillation (Meng et al., 2023) run two steps using the DDIM sampler (equation 6).

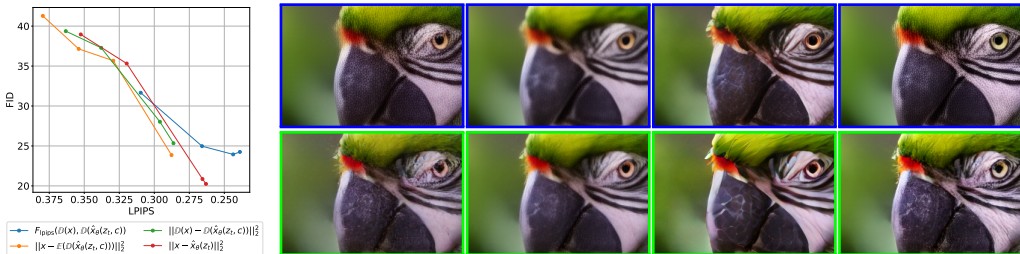

Figure 3: Sampled results between distilled models learned with alternative conditional guidance. Left curves shows the quantitative performance between the LPIPS and FID in $\{1, 2, 4, 8\}$ steps. Right part show the visual results where each result comes from the 1 sampling step (top) or 4 sampling steps (bottom). The distance function from the left to right is $\|\mathbf{x} - \mathbb{E}(\mathbb{D}(\hat{\mathbf{x}}_\theta(\mathbf{z}_t, c)))\|_2^2$, $\|\mathbb{D}(\mathbf{x}) - \mathbb{D}(\hat{\mathbf{x}}_\theta(\mathbf{z}_t, c))\|_2^2$, $F_{\text{lpips}}(\mathbb{D}(\mathbf{x}), \mathbb{D}(\hat{\mathbf{x}}_\theta(\mathbf{z}_t, c))$, and our default $\|\mathbf{x} - \hat{\mathbf{x}}_\theta(\mathbf{z}_t)\|_2^2$, respectively.

Here, we propose an alternative prediction for $\hat{\mathbf{z}}_s$. Our prediction depends on the signal prediction $\hat{\mathbf{x}}_\theta(\mathbf{z}_t, c, t)$ from the adapted diffusion model, and the original random noise $\epsilon$ used when sampling $\mathbf{z}_t$. We dubbed this *partial real-value predictor (PREv-predictor)*, and as the reader may see, it consists of replacing the noise prediction in the DDIM sampler (equation 6) by the real noise $\epsilon$,

$$\mathbf{z}_t = \alpha_t \mathbf{x} + \sigma_t \epsilon, \epsilon \sim \mathcal{N}(0, \mathbf{I}), \text{and } \hat{\mathbf{z}}_s = \alpha_s \hat{\mathbf{x}}_\theta(\mathbf{z}_t, c, t) + \sigma_s \epsilon. \tag{12}$$

Specifically, we first sample $\epsilon \sim \mathcal{N}(0, \mathbf{I})$ for generating $\mathbf{z}_t$, and then use the same noise $\epsilon$ in $\hat{\mathbf{z}}_s$.

The generated $\hat{\mathbf{z}}_s$ not only depends on the conditional diffusion model prediction $\hat{\mathbf{x}}_\theta(\mathbf{z}_t, c, t)$, but also on the stochastic noise component. As we show in Figure 5, this leads to better performance than directly using the prediction from DDIM (equation 6).

### 4.3 CONDITIONAL GUIDANCE

To finetune the adapted diffusion model with the new conditional data, our conditional diffusion distillation loss in equation 11 penalizes the difference between the predicted signal $\hat{\mathbf{x}}_\theta(\mathbf{z}_t, c, t)$ and the corresponding image $\mathbf{x}$ with a distance function $d_\mathbf{x}(\cdot, \cdot)$ for distillation learning.

Here we investigate the impact of the distance function $d_\mathbf{x}(\cdot, \cdot)$ in the conditional guidance. According to both qualitative and quantitative results, shown in Figure 3, different distance functions lead to different behaviours when doing multi-step sampling (inference). If $d_\mathbf{x} = \|\cdot\|^2$ in the pixel space or the encoded space, *i.e.*, $\|\mathbf{x} - \mathbb{E}(\mathbb{D}(\hat{\mathbf{x}}_\theta(\mathbf{z}_t, c, t)))\|_2^2$ and $\|\mathbb{D}(\mathbf{x}) - \mathbb{D}(\hat{\mathbf{x}}_\theta(\mathbf{z}_t, c, t))\|_2^2$, multi-step sampling leads to more smooth and blurry results. If instead we adopt a perceptual distance in the pixel space, *i.e.*, $\mathcal{F}_{\text{lpips}}(\mathbb{D}(\mathbf{x}), \mathbb{D}(\hat{\mathbf{x}}_\theta(\mathbf{z}_t, c, t)))$, the iterative refinement in the multi-step sampling leads to over-saturated results. Overall, by default we adopted the $\ell_2$ distance in the latent space since it leads to better visual quality and achieve the optimal FID with 4 sampling steps in Figure 3.

### 4.4 PARAMETER-EFFICIENT CONDITIONAL DISTILLATION

Our method offers the flexibility to selectively update parameters pertinent to distillation and conditional finetuning, leaving the remaining parameters frozen. This leads us to introduce a new fashion of parameter-efficient conditional distillation, aiming at unifying the distillation process across commonly-used parameter-efficient diffusion model finetuning, including ControlNet (Zhang & Agrawala, 2023), T2I-Adapter (Mou et al., 2023), etc.

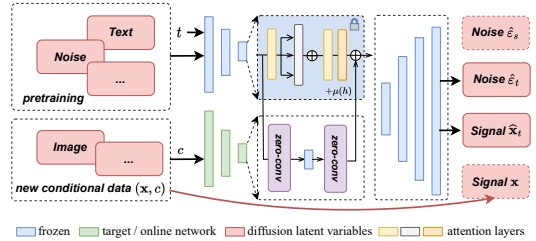

Figure 4: Network architecture illustration of our parameter-efficient conditional distillation framework.

We highlight the ControlNet architecture illustrated in Figure 4 as an example. This model duplicates the encoder part of the denoising network, highlighted in the green blocks, as the condition-related parameters. Similar to Algorithm 1, our

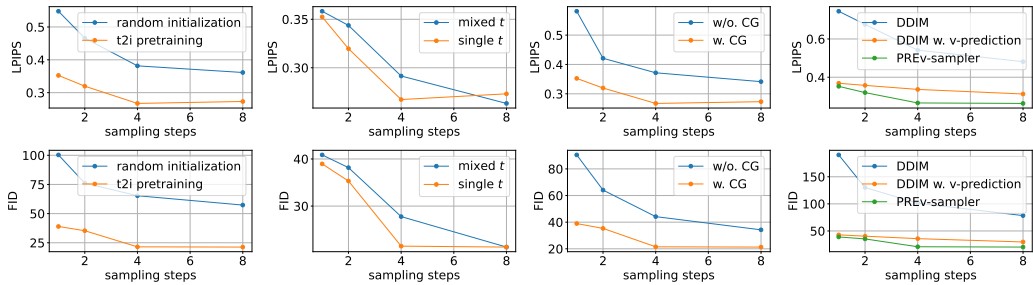

Figure 5: Ablations between different alternative settings of our method, where each point corresponds to the result under different sampling steps (*i.e.*, $\{1, 2, 4, 8\}$).

distillation objective is to minimize the noise prediction, but instead, this prediction comes from the combination of the frozen denoising network and the learnable conditional adapter.

# 5 EXPERIMENTS

We demonstrate the efficacy of our method on representative conditional generation tasks, including, real-world super-resolution (Wang et al., 2022), depth-to-image generation (Zhang & Agrawala, 2023), and instructed image editing (Brooks et al., 2023). We utilize a pre-trained text-to-image generation model[3] and conduct conditional distillation directly from the unconditional model. Prior to presenting our results, we first detail the ablations of each hyperparameter in our method.

## 5.1 ABLATIONS

Here we compare the performance of the aforementioned designs in our conditional distillation framework. Specifically we focus on the representative conditional generation task *i.e.*, real-world super-resolution (Wang et al., 2022) that conditions on the low-resolution, noisy, blurry images.

**Pretraining.** To validate the effectiveness of leveraging pretraining in our model, we compare the results of random initialization with initialization from the pre-trained text-to-image model. As shown in Figure 5, our method outperforms the random initialized counterpart by a large margin, thereby confirming that our strategy indeed utilizes the advantages of pretraining during distillation instead of simply learning from scratch.

**Sampling of $\mathbf{z}_t$.** We empirically show that the way of sampling $\mathbf{z}_t$ plays a crucial role in the distillation learning process. Compared with the previous protocol (Salimans & Ho, 2022; Meng et al., 2023) that samples $\mathbf{z}_t$ in different time $t$ in a single batch, we show that using a consistent time $t$ across different samples in a single batch leads to a better performance. As the comparisons shown in Figure 5, the model trained with a single time $t$ (in a single batch) achieves better performance in both the visual quality (*i.e.*, FID) and the accuracy (*i.e.*, LPIPS) when the number of evaluations is increasing during inference. As our joint-learning is challenging, we believe this simplified sampling protocol helps to simplify the learning by learning on images in the same noise level.

**Conditional guidance.** In order to demonstrate the importance of our proposed conditional guidance (CG) for distillation, which is claimed to be capable of regularizing the distillation process during training, we conduct comparisons between the setting of using the conditional guidance as $r = \|\mathbf{x} - \hat{\mathbf{x}}_\theta(\mathbf{z}_t, c)\|_2^2$ and not using as $r = 0$. As the result shown in Figure 5, the conditional guidance improves both the fidelity of the generated results and visual quality. We further observed that the distillation process will converge toward over-saturated direction without CG, which thus lower the FID metric. In contrast, our model can avoid such a local minimum because our learning is lower bounded by the guidance loss, which can be seen as a typical diffusion loss.

---

[3]We base our work on a version of Latent Diffusion Model trained on internal data sources.

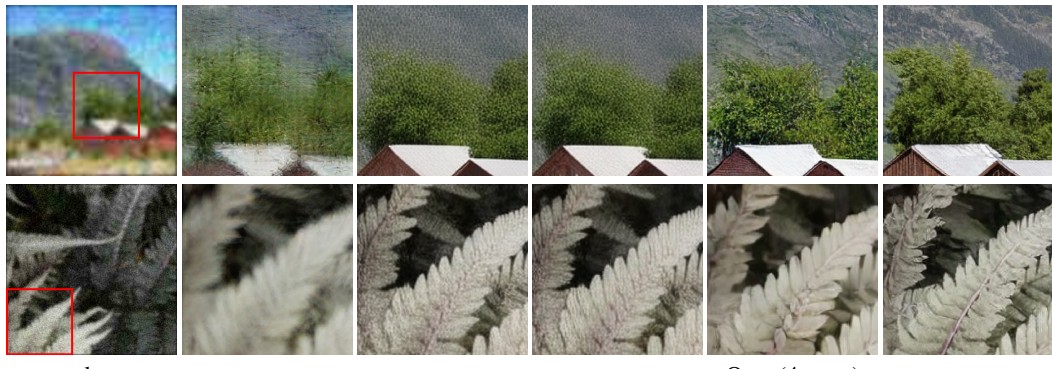

lr    StableSR (200 steps)    LDM-SR (4 steps)    GuidedDistill (4 steps)    Ours (4 steps)    LDM-SR (200 steps)

Figure 6: Visual comparisons of various diffusion-based methods and ours on the super-resolution data that has noise, compression, and blur degraded images. Compared with the other methods, our distilled model achieves the best visual quality by using less sampling time.

**Prediction of $\hat{\mathbf{z}}_s$.** In order to demonstrate the effects of our proposed PREv-predictor that uses the initial noise $\epsilon$, we conduct comparisons between the deterministic predictor (*i.e.*, equation 6) used by previous distillation methods (Salimans & Ho, 2022), the adapted DDIM predictor in the velocity model (Salimans & Ho, 2022), and ours. As shown in Figure 5, both the derived DDIM with v-prediction and our proposed PREv-predictor benefit to the distillation, while the previous DDIM predictor that solely depends on the pretrained model without using sampled noise $\epsilon$ fails at the conditional distillation learning. Moreover, we empirically find that our PREv-predictor that utilizes the sampled $\epsilon$ can achieve slightly better performance than the DDIM with v-prediction.

## 5.2 RESULTS

**Real-world super-resolution.** We demonstrate our method on the challenging real-world super-resolution task, where the degradation is simulated using the Real-ESRGAN pipeline (Wang et al., 2021). We compare our distilled model against the fine-tuned latent diffusion-based model (LDM-SR) (Rombach et al., 2022) with different sampling steps, and the distilled LDM by using guided-distillation (GD) (Meng et al., 2023) in a distillation-first fashion and consistency models (CM) (Song et al., 2023) in a conditional finetuning-first fashion. We alternatively compare the recent fast ODE solver including DPM-Solver Lu et al. (2022a) and DPM-Solver++ Lu et al. (2022b). Moreover, we also include our parameter-efficient distillation, where only the conditional adapter is trained.

| Method | Params | Steps | FID | LPIPS |
|---|---|---|---|---|
| Real-ESRGAN | 16.6M | 1 | 37.64 | 0.3112 |
| StableSR | 865M | 200 | 24.44 | 0.3114 |
| LDM-SR | 1.22B | 4 | 30.99 | 0.3070 |
| DPM Solver | 1.22B | 4 | 30.12 | 0.3077 |
| DPM Solver++ | 1.22B | 4 | 30.03 | 0.3073 |
| CM | 1.22B | 4 | 30.63 | 0.3066 |
| GD | 1.22B | 4 | 27.81 | 0.3172 |
| **Ours*** | 364M | 4 | *25.21* | *0.2941* |
| **Ours** | 1.22B | 4 | **19.64** | **0.2656** |
| LDM-SR | 1.22B | 200 | 18.63 | 0.2551 |

Table 1: Quantitative performance comparisons on real-world super-resolution.

The quantitative performance is shown in Table 1. The results demonstrate that our distilled method inherits the performance superiority from the pretraining. It even achieves better results than the fine-tuned diffusion models that costs $50\times$ more sampling time, which benefits from the joint distillation-finetuning optimization. Compared with the distilled model by applying the guided-distillation, our model outperforms it both quantitatively and in visual quality as shown in Figure 6.

**Depth-to-image generation.** In order to demonstrate the generality of our method on less informative conditions, we apply our method in depth-to-image generation. The task is usually conducted in parameter-efficient diffusion model finetuning (Mou et al., 2023; Zhang & Agrawala, 2023), which can demonstrate the capability of utilizing text-to-image generation priors. As Figure 7 illustrated, our distilled model from the unconditional pretraining can effectively utilize the less informative conditions and generate matched images with more details, while the fine-tuned model can hardly generated reasonable results in the same sampling steps.

**Instructed image editing.** To demonstrate our conditional distillation capability on text-to-image generation, here we apply our method on text-instructed image editing data (Brooks et al., 2023)

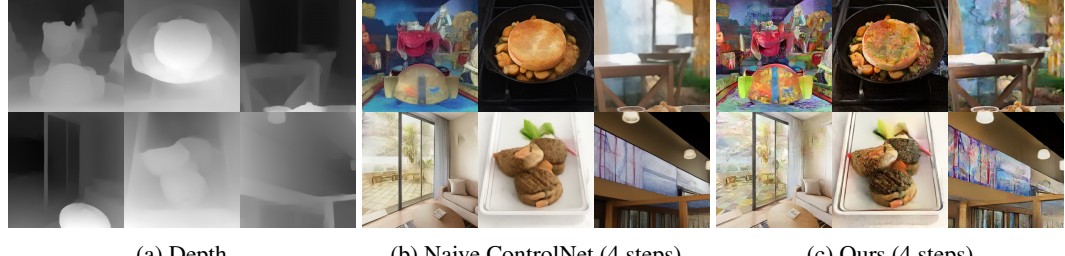

(a) Depth     (b) Naive ControlNet (4 steps)     (c) Ours (4 steps)

Figure 7: Samples generated according to the depth image (left) from ControlNet sampled in 200 steps (middle left), ControlNet sampled in 4 steps (middle right), and our distilled ControlNet from the unconditional pretraining sampled in 4 steps (right).

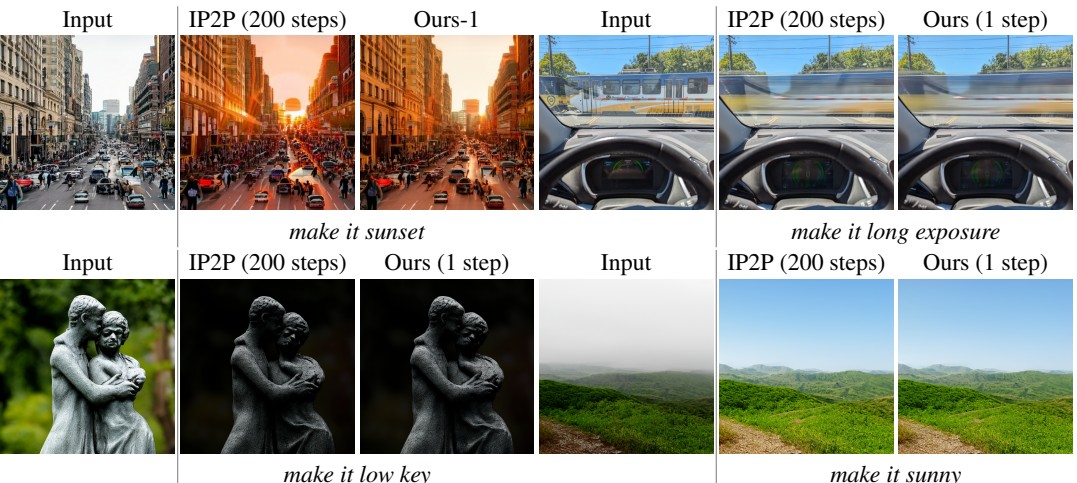

Figure 8: Generated edited image according to the input image and the instruction (bottom) from Instructed Pix2Pix (IP2P) sampled in 200 steps and ours sampled in 1 step.

and compare our conditional distilled model with the InstructPix2Pix (IP2P) model. As the results shown in Figure 8, our single-step sampling result can achieve comparable visual quality to 200 steps of the IP2P model. We experimentally find only small visual difference between the results from our single-step sampling and the 200 steps sampling. We believe this suggests that the effect of the conditional guidance on distillation correlates with the similarity between the conditions and the target data, further demonstrating the effectiveness of our conditional diffusion distillation.

## 6 CONCLUSION

We introduce a new framework for distilling an unconditional diffusion model into a conditional one that allows sampling with very few steps. To the best of our knowledge, this is the first method that distills the conditional diffusion model from the unconditional pretraining in a single stage. Compared with previous two-stage distillation and finetuning techniques, our method leads to better quality given the same number of (very few) sampling steps. Our method also enables a new parameter-efficient distillation that allows different distilled models, trained for different tasks, to share most of their parameters. Only a few additional parameters are needed for each different conditional generation task. We believe the method can serve as a strong practical approach for accelerating large-scale conditional diffusion models.

**Limitations.** We have shown image conditions benefit our distillation learning. However, the distillation learning depends on the adapter architecture that takes conditions, and it is difficult to reduce the inference latency introduced by the adapter network in our current framework. As a future work, we would like to explore lightweight network architectures (Li et al., 2023) in our distillation technique to further reduce the inference latency.

**Reproducibility statement.** Our implementation can be reproduced according to the Algorithm 1. In Section 5.1 and Section 4.3, we show the effects of using different hypereparameters, providing references to the reader who wants to implement the method for customized applications.

**Ethics statement.** The diffusion distillation technique introduce in this work holds the promise of significantly enhancing the practicality of diffusion models in everyday applications such as consumer photography and artistic creation. While we are excited about the possibilities this model offers, we are also acutely aware of the possible risks and challenges associated with its deployment. Our model's ability to generate realistic scenes could be misused for generating deceptive content. We encourage the research community and practitioners to prioritize privacy-preserving practices when using our method. Additionally, we recommend that readers refer to the work by Rostamzadeh et al. (2021) for a thorough examination of ethics in generating visual content.

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
