# A PROOFS

## A.1 NOTATIONS

We use $\hat{\mathbf{v}}_\theta(\cdot, \cdot)$ to denote a pre-trained diffusion model that learns the unconditional data distribution $\mathbf{x} \sim p_{\text{data}}$ with parameters $\theta$. The signal prediction and the noise prediction transformed by equation 7 are denoted by $\hat{\mathbf{x}}_\theta(\cdot, \cdot)$ and $\hat{\epsilon}_\theta(\cdot, \cdot)$, and they share the same parameters $\theta$ with $\hat{\mathbf{v}}_\theta(\cdot, \cdot)$.

## A.2 SELF-CONSISTENCY IN NOISE PREDICTION

**Remark.** *If a diffusion model, parameterized by $\hat{\mathbf{v}}_\theta(\mathbf{z}_t, t)$, satisfies the self-consistency property on the noise prediction $\hat{\epsilon}_\theta(\mathbf{z}_t, t) = \alpha_t \hat{\mathbf{v}}_\theta(\mathbf{z}_t, t) + \sigma_t \mathbf{z}_t$, then it also satisfies the self-consistency property on the signal prediction $\hat{\mathbf{x}}_\theta(\mathbf{z}_t, t) = \alpha_t \mathbf{z}_t - \sigma_t \hat{\mathbf{v}}_\theta(\mathbf{z}_t, t)$.*

*Proof.* The diffusion model that satisfies the self-consistency in the noise prediction implies:

$$
\begin{aligned}
\hat{\epsilon}_\theta(\mathbf{z}_{t'}, t') &= \hat{\epsilon}_\theta(\mathbf{z}_t, t), \\
\alpha_{t'} \hat{\mathbf{v}}_\theta(\mathbf{z}_{t'}, t') + \sigma_{t'} \mathbf{z}_{t'} &= \alpha_t \hat{\mathbf{v}}_\theta(\mathbf{z}_t, t) + \sigma_t \mathbf{z}_t, \\
\hat{\mathbf{v}}_\theta(\mathbf{z}_{t'}, t') &= \frac{\alpha_t \hat{\mathbf{v}}_\theta(\mathbf{z}_t, t) + \sigma_t \mathbf{z}_t - \sigma_{t'} \mathbf{z}_{t'}}{\alpha_{t'}},
\end{aligned}
\tag{13}
$$

Based on the above equivalence, the transformation between the signal prediction $\mathbf{x}_\theta(\mathbf{z}_{t'}, t')$ and $\mathbf{x}_\theta(\mathbf{z}_t, t)$ by using the update ruler in equation 6 and the reparameterization trick is:

$$
\begin{aligned}
\mathbf{x}_\theta(\mathbf{z}_{t'}, t') &= \alpha_{t'} \mathbf{z}_{t'} - \sigma_{t'} \hat{\mathbf{v}}_\theta(\mathbf{z}_{t'}, t') \\
&= \alpha_{t'} \mathbf{z}_{t'} - \sigma_{t'} \frac{\alpha_t \hat{\mathbf{v}}_\theta(\mathbf{z}_t, t) + \sigma_t \mathbf{z}_t - \sigma_{t'} \mathbf{z}_{t'}}{\alpha_{t'}} && \text{// integrating equation 13} \\
&= \frac{\alpha_{t'}^2 \mathbf{z}_{t'} - \sigma_{t'} \alpha_t \hat{\mathbf{v}}_\theta(\mathbf{z}_t, t) - \sigma_{t'} \sigma_t \mathbf{z}_t + \sigma_{t'}^2 \mathbf{z}_{t'}}{\alpha_{t'}} \\
&= \frac{(1 - \sigma_{t'}^2) \mathbf{z}_{t'} - \sigma_{t'} \alpha_t \hat{\mathbf{v}}_\theta(\mathbf{z}_t, t) - \sigma_{t'} \sigma_t \mathbf{z}_t + \sigma_{t'}^2 \mathbf{z}_{t'}}{\alpha_{t'}} \\
&= \frac{\mathbf{z}_{t'} - \sigma_{t'} (\alpha_t \hat{\mathbf{v}}_\theta(\mathbf{z}_t, t) + \sigma_t \mathbf{z}_t)}{\alpha_{t'}} \\
&= \frac{\mathbf{z}_{t'} - \sigma_{t'} (\hat{\epsilon}_\theta(\mathbf{z}_t, t))}{\alpha_{t'}} && \text{// transformed with equation 7} \\
&= \frac{\alpha_{t'} \mathbf{x}_\theta(\mathbf{z}_t, t) + \sigma_{t'} \hat{\epsilon}_\theta(\mathbf{z}_t, t) - \sigma_{t'} (\hat{\epsilon}_\theta(\mathbf{z}_t, t))}{\alpha_{t'}} && \text{// update ruler equation 8 of DDIM} \\
&= \mathbf{x}_\theta(\mathbf{z}_t, t).
\end{aligned}
$$

The derived equivalence shows that enforcing the self-consistency in the noise prediction, which is implemented by learning to minimize our distillation loss in equation 11, enforces the self-consistency in the signal prediction and can distill the pre-trained diffusion model.

# B SAMPLING PROCESS VISUALIZATION

In order to provide a comprehensive understanding about the sampling process of our distilled model, as well as the difference between ours and the finetuned conditional diffusion model, here we visualize their predicted clean image $\hat{\mathbf{x}}_0$ at each sampling steps in equation 7.

As the results shown in Figure 9, we can find that our method constantly adds more details into the predicted $\hat{\mathbf{x}}_0$ when samples more steps. In contrast, such a constantly refinement is less visible in the results of the finetuned undistilled model. The different demonstrate that our method indeed can reduce the sampling time by learning to replicate the iterative refinement effects.

*sampling steps $0 \leftarrow T$ with finetuned model*

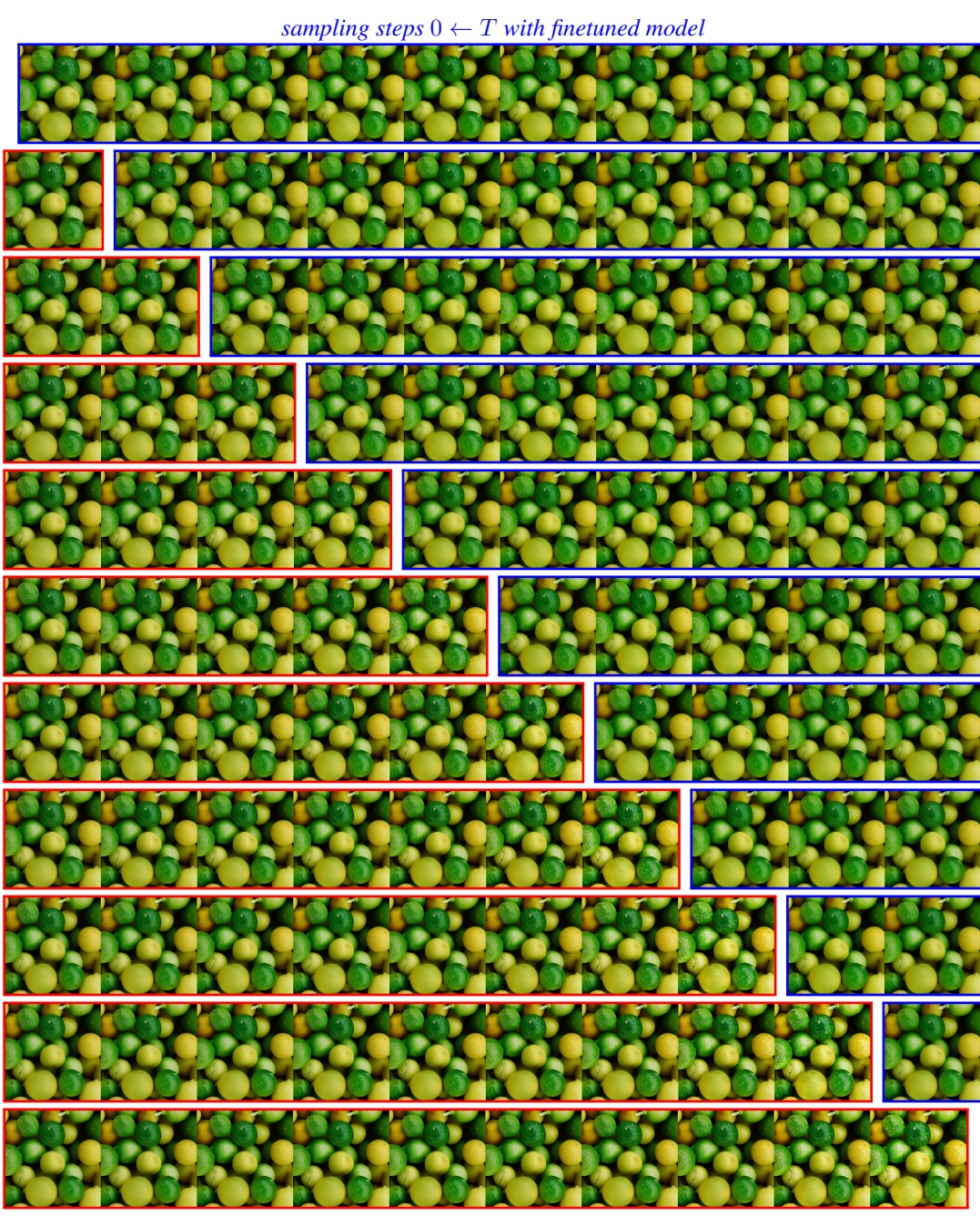

*sampling steps $T \rightarrow 0$ with our conditional distilled model*

Figure 9: Sampling process visualization of the distilled model by using our conditional diffusion distillation and the finetuned conditional diffusion model. The results belong to the same row come from the predicted $\hat{x}_0$ at different time of the same sampling process, while different row denotes different sampling process that uses different the total number of the sampling time, which are increased from $T = 0$ into $T = 10$ and decreased from $T = 10$ into $T = 0$, respectively.

## C   ADDITIONAL RESULTS

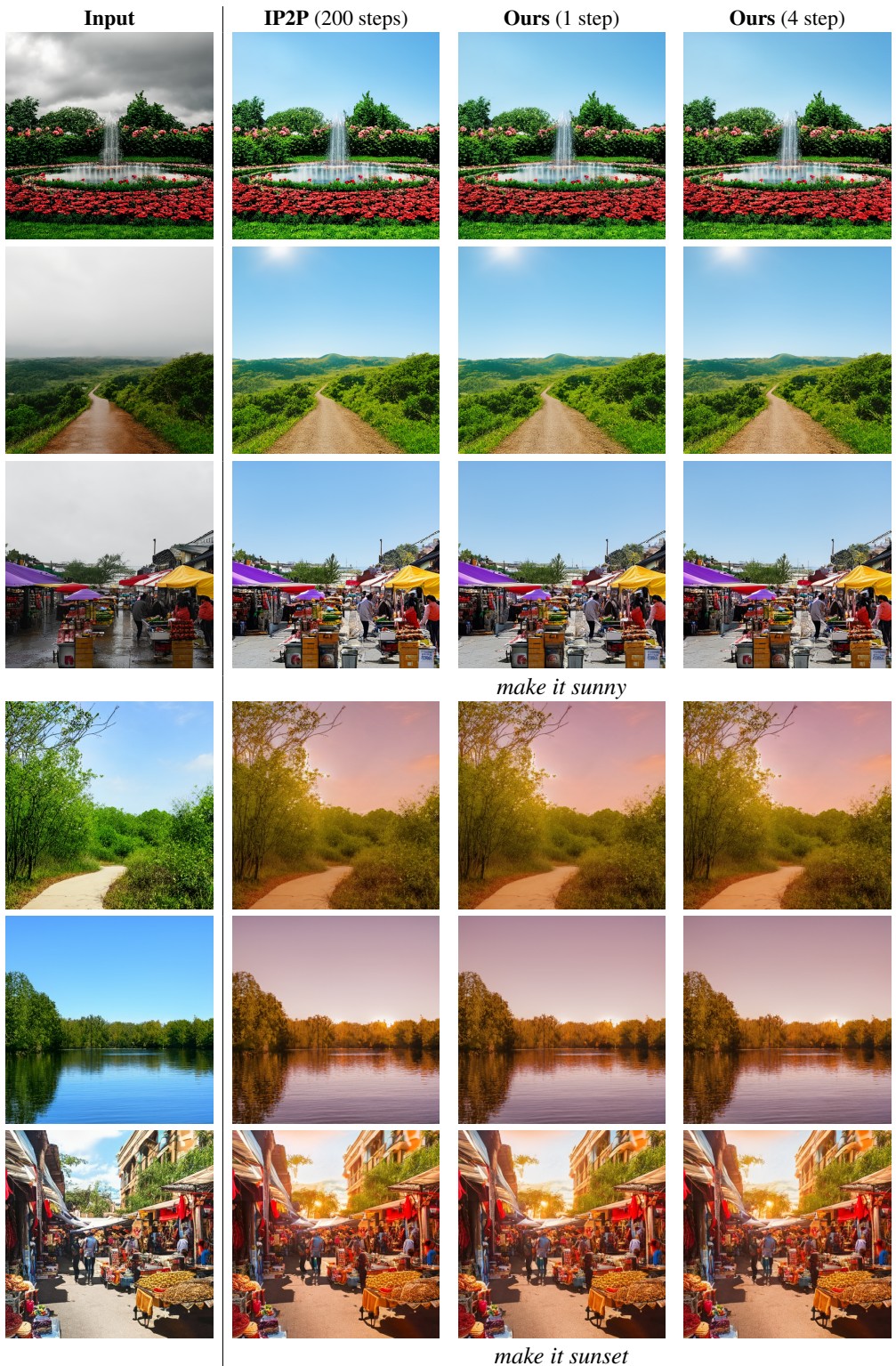

Figure 10: Visual comparisons with the IP2P model and our conditional distilled model.

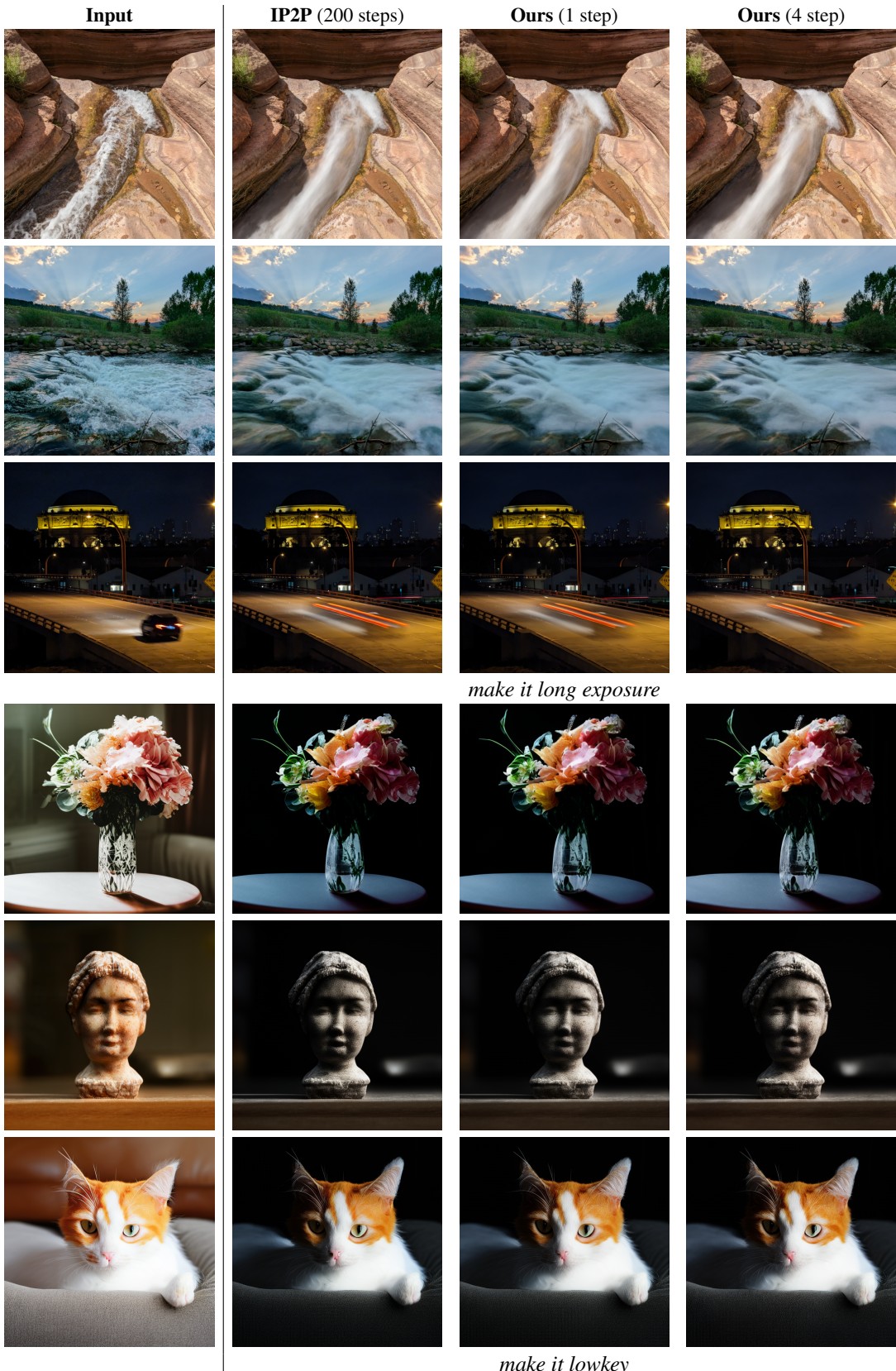

Figure 11: Visual comparisons with the IP2P model and our conditional distilled model.

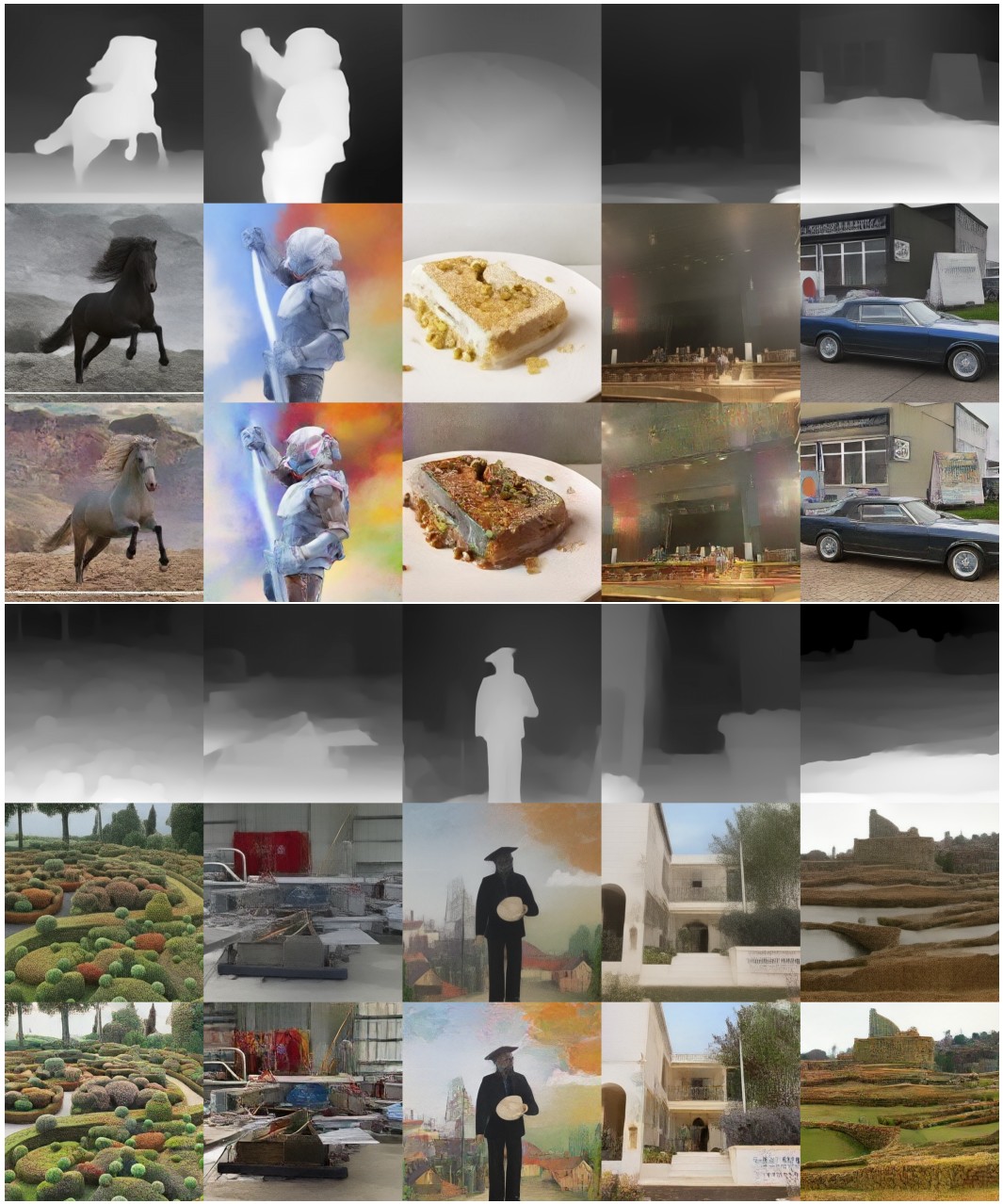

Figure 12: Visual comparisons of depth to image generation with the native ControlNet (central row of each item) and our conditional distilled model (bottom row of each item) in 4 sampling steps.

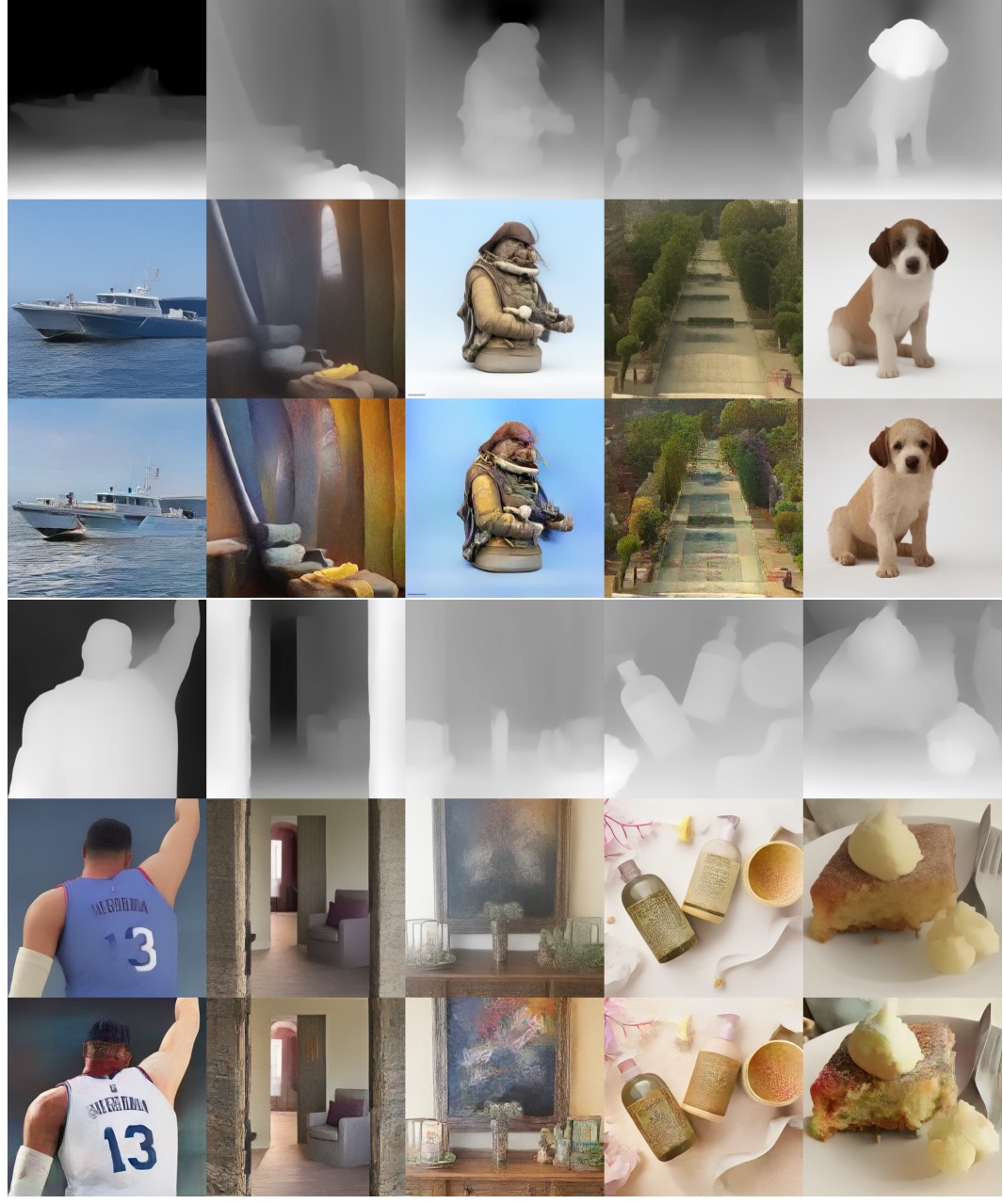

Figure 13: Visual comparisons of depth to image generation with the native ControlNet (central row of each item) and our conditional distilled model (bottom row of each item) in 4 sampling steps.

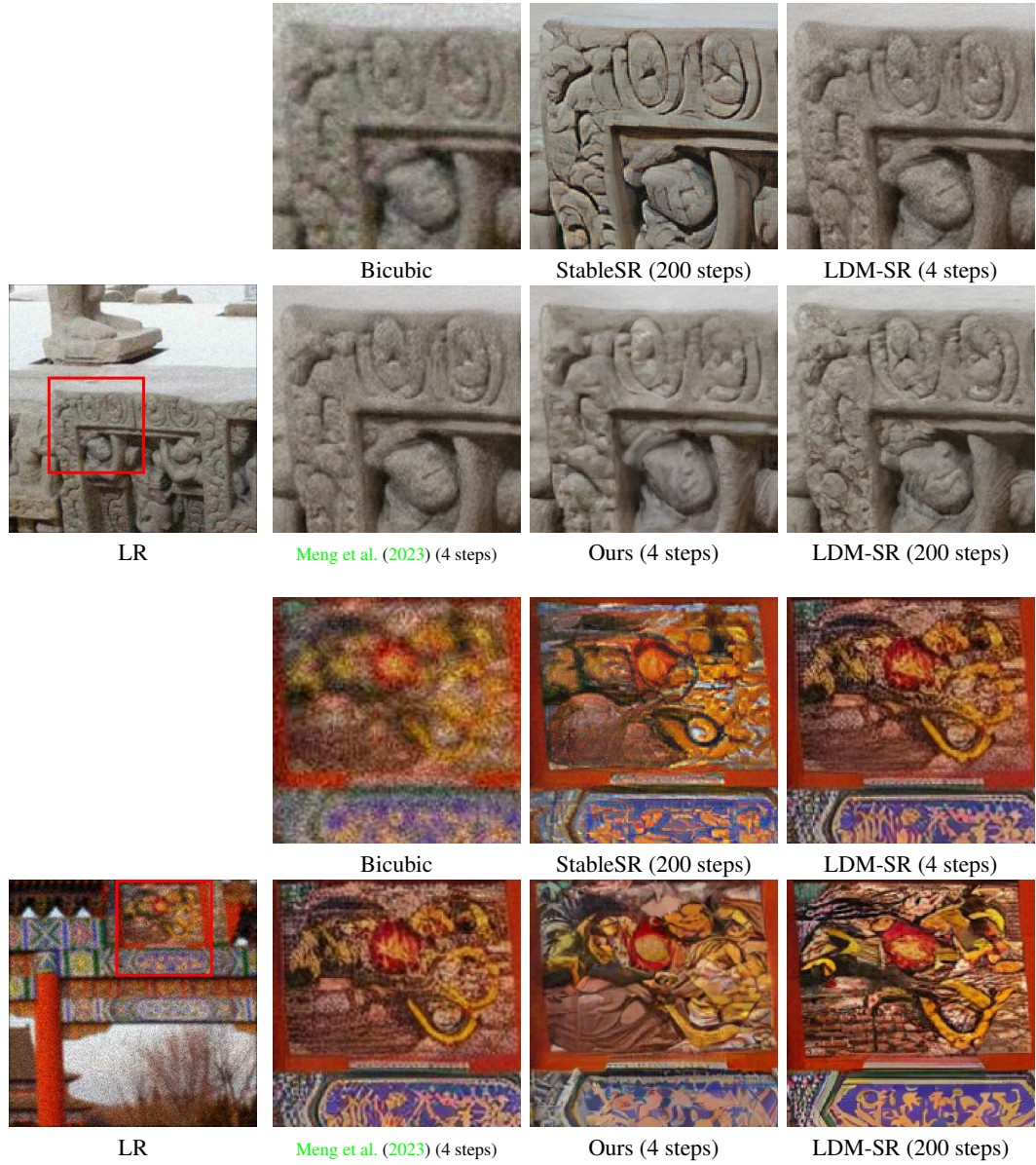

Figure 14: Visual comparisons of various diffusion-based methods on the simulated real-world super-resolution benchmark. The input of all methods is a 'Bicubic'-upsampled image.

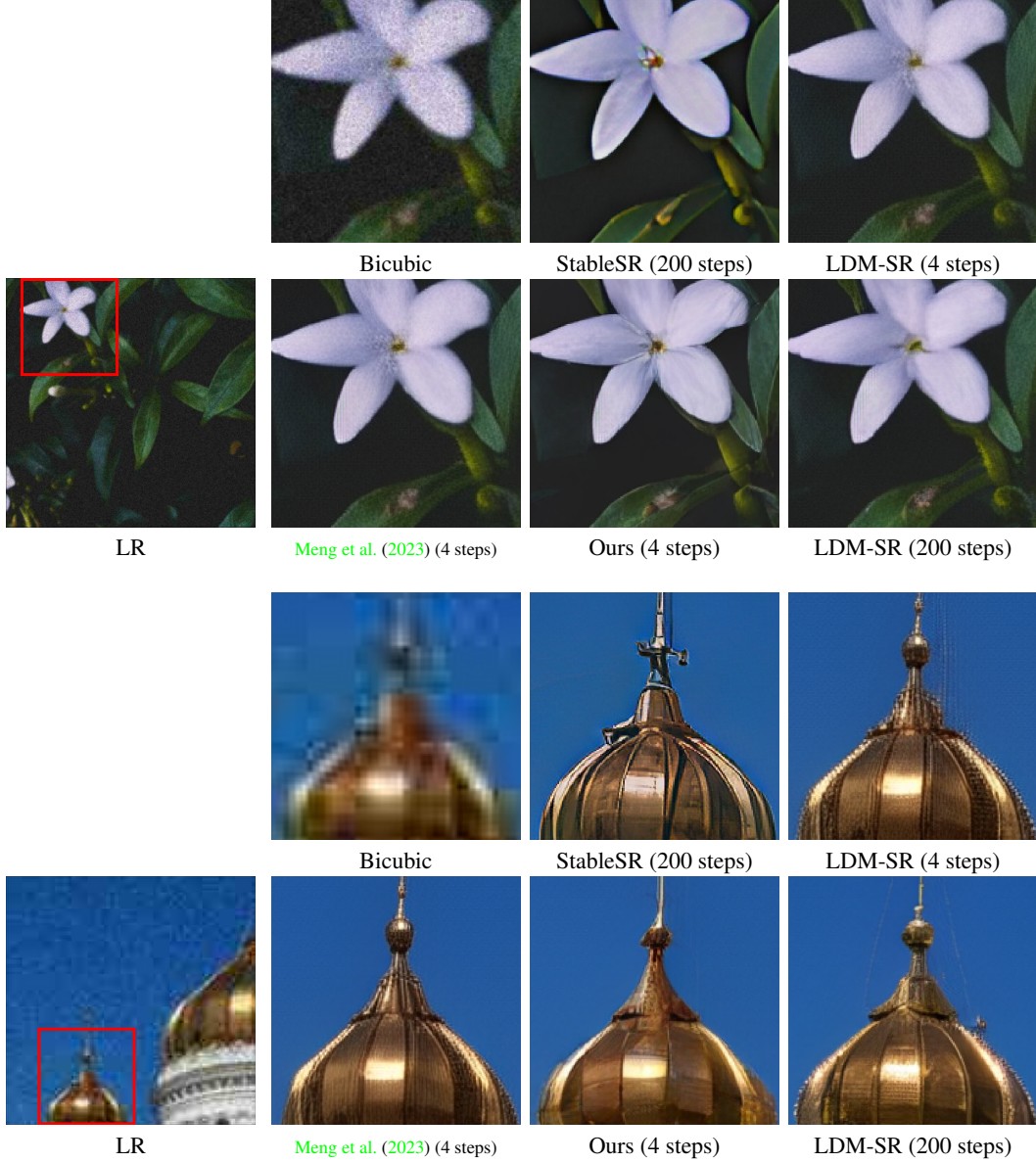

Figure 15: Visual comparisons of various diffusion-based methods on the simulated real-world super-resolution benchmark. The input of all methods is a 'Bicubic'-upsampled image.