# OpenReview forum: "Conditional Diffusion Distillation"
_ICLR.cc/2024/Conference — ICLR 2024 Conference Withdrawn Submission_

### Official Review · Reviewer_ugTK · 2023-10-19

**Soundness:** 2 fair
**Presentation:** 1 poor
**Contribution:** 2 fair
**Rating:** 3
**Confidence:** 3

**Summary:**

This paper seeks to accelerate the inference process of diffusion models by supplementing the diffusion priors with image conditions. Essentially by providing additional images prior to base conditional diffusion models.

**Strengths:**

The motivation to accelerate the diffusion inference process is reasonable, the experiments show improvements over existing distillation methods.

**Weaknesses:**

I hope the authors don’t take the comments personal, but I don’t think the paper is yet ready to be published based on the following reasons.

- First, I think the clarity of the paper presentation can be largely improved, I was having a very hard time understanding the paper and trying to extract the technical designs/contributions in my first reading, even though I believe myself to be quite familiar with the diffusion models.

- In the introduction, the authors state “a two stage distillation procedure can be used for distilling conditional diffusion models - either distillation-first or conditioning finetuning first.” This is difficult for readers outside the distillation field to understand, a high-level conceptual clarification is at least expected following this sentence. What is distillation first, and what is conditioning finetuning first? What are their differences?

- In Fig. 1, the authors include some qualitative examples from the proposed distillation method. Since the diffusion field is getting very crowded and every paper submission shows good qualitative results in the teaser figure, it is more helpful for authors to include a comparison with other existing distillation methods, instead of just showing a few examples of the proposed method (in other words, Fig.1 is not very informative).

- A side note on the claim ``These two procedures offer different advantages in terms of cross-task flexibility and learning difficulty, but their generated results are generally better than those of the undistilled conditional diffusion model when given the same sampling time’’, I don’t think it is a fair comparison, and of course, the distilled conditional model should perform better than undistilled ones given the same sampling time, this is what distillation methods designed for?

- I also had a difficult time seeing the connections between the presented background and the scope of this paper. Or I guess another way to put my question is: why Eq. (1) - (8) should be included in the main paper?

- Why T2I models such as StableDiffusion and Imagen are considered unconditional models as stated in footnote 2?

- Is $\mu$ in Eq. (10) a learnable parameter? But has not been discussed in main experiments or ablation studies?

- A suggestion on Fig.2, I think the current Fig.2 is too small to read as this is your main methodology figure…

- The computational resources and the time cost are not reported and discussed?

- Many references in this manuscript have the format of their ArXiv version (such as [a] [b] [c]), while this is a minor suggestion, I believe it should be the authors’ responsibility to update the bibliography with their final publication venues.

[a] Denoising diffusion implicit models.

[b] Score-based generative modeling through stochastic differential equations.

[c] Sdedit: Guided image synthesis and editing with stochastic differential equations.

**Questions:**

Please see the Weaknesses for details.

---

### Official Review · Reviewer_9yxs · 2023-11-01

**Soundness:** 2 fair
**Presentation:** 2 fair
**Contribution:** 3 good
**Rating:** 3
**Confidence:** 4

**Summary:**

The paper proposes a method for jointly distilling and fine-tuning a large diffusion model for conditional sampling. The authors introduce a series of architectural and algorithmic improvements to train fast, conditional samplers starting from a pre-trained diffusion generative model.

**Strengths:**

- The authors successfully combine diffusion model distillation with conditioning methodologies to present a novel approach to joint distillation and conditioning. This could be a significant contribution to the diffusion model community as the need for personalized models is always increasing.

- Strong qualitative experimental results. The authors demonstrate impressive distillation capabilities by showing that they can significantly reduce the number of function evaluations without significant loss in the quality of the images. This is exemplified by three distinct tasks.

**Weaknesses:**

- The writing of the paper is not clear. The authors briefly discuss each of their ideas but there is no single point that stands out as the main contribution. Sections 4.1 and 4.4 seem to present the same conditional architecture. It would help if the authors clearly separated their contributions on the architectural and algorithmic side.

- The authors present quantitative results only on the super-resolution task. It would be important to also show some quantitative results on a second task. For instance, CLIP similarity between original and edited images and CLIP similarity between the editing caption and the edited image. FID on a depth-to-image dataset could also help.

- The related works section could be a bit clearer. There needs to be a grouping of the previous methods (full inference vs step-by-step distillation for example) to contrast between the different approaches and help understand where the proposed approach stands. In its current form, it is difficult to understand what the advantages/disadvantages of existing works are and where the authors aim to make a contribution.

**Questions:**

- Can the authors elaborate on the proposed "PREv-predictor"? Is there an intuition on why using the same noise leads to better results?

- How does the proposed approach compare to fine-tuning the model and then distilling it? Is performance sacrificed by doing both in a single stage?

- What are the data requirements for this distillation approach? Do you need a dataset of comparable size to ControlNet? Previous distillation approaches relied on having access to the full dataset or a large synthetic pre-generated set.

---

### Official Review · Reviewer_KX2e · 2023-11-01

**Soundness:** 3 good
**Presentation:** 3 good
**Contribution:** 2 fair
**Rating:** 5
**Confidence:** 3

**Summary:**

The work proposes a scheme to distill information from an unconditional diffusion model to a conditional diffusion model.

**Strengths:**

The writing of the paper is good and the proposes have some improvements over the baselines

**Weaknesses:**

The motivation of the work is the main weaknesses:

1. What is the main aim of the work? Is the work aiming to achieve a conditional model from an unconditional or is the main objective is to distill from large sampling timesteps to a small number of timesteps? It is quite confusing to the readers whether they need to use this one for fast diffusion or use this one to add conditional information. It is better to focus on one aspect, and the other is an extension as a plus.
2. If it is about achieving conditional information for an unconditional diffusion model, there is a need to compare the proposed method with a finetuning scheme with parameters initialized as unconditional diffusion model.
3. The authors did not discuss in which case the work should be utilized. This information should be added into the manuscripts to let the readers imagine about the use cases.
4. Based on the Figure 6, it is not easy to tell the proposed method is better or not.

**Questions:**

See the weaknesses. Will adapt the scores after rebuttals if the concerns are solved.

**Details Of Ethics Concerns:**

N.A

---

### Official Review · Reviewer_UAJi · 2023-11-06

**Soundness:** 2 fair
**Presentation:** 3 good
**Contribution:** 2 fair
**Rating:** 3
**Confidence:** 4

**Summary:**

The authors present a method for creating a conditional distilled diffusion model directly from an unconditional pre-trained diffusion model. This method bypasses the usual step of first fine-tuning a conditional model. The approach draws on consistency models, and it uses a mix of consistency and diffusion losses. The authors have also tweaked the way training data points are sampled to boost the model's performance. Furthermore, they suggest a way to make the distilled model use fewer parameters, taking cues from ControlNet. The paper shows that this new distillation technique performs well when compared to both fast diffusion samplers and earlier diffusion distillation methods.

**Strengths:**

* The method presented in this paper is original and well motivated. It introduces a way to skip the usual finetuning step needed for converting unconditional diffusion models to fast conditional diffusion samplers, allowing for the creation of distilled conditional models directly. This simplification of the process is a useful improvement. The way the paper combines consistency distillation with diffusion loss is both new and intuitive. The concept of using a partial real-value predictor is also a compelling addition that seems to enhance distillation performance.

* The paper is well-structured and easy to read.

* The issue that this paper addresses is significant. There's a need for methods that can quickly and effectively adapt powerful, all-purpose diffusion models to particular areas.

**Weaknesses:**

The paper requires more rigor in both its theoretical explanations and experimental validation.

Theoretical concerns include:

- The compatibility between the conditional guidance term $d_x(\cdot, \cdot)$ and the consistency loss term $d_\epsilon(\cdot, \cdot)$ is not well-explained. The paper describes the guidance as analogous to the classical denoising score matching loss found in diffusion models, where the ideal model predicts the average of all possible noiseless images that could have led to the given noisy image. However, the consistency loss term aims for a model that predicts a singular clean image for a noisy input, as directed by the probability flow ODE trajectories. These objectives seem to conflict with each other.

- The justification for the improved performance due to the partial real-value predictor is not clear. Despite its practical effectiveness, the paper does not provide a convincing explanation for why this adjustment is beneficial.

Experimentally, there are several issues:

- The ablation study lacks a critical comparison. It does not include a baseline where an unconditional diffusion model is first refined using the suggested efficient architectural changes, followed by either consistency distillation or progressive distillation. It would be valuable to determine if the proposed method would outperform these conventional baselines under the same computational constraints. Comparing to a model initiated with random parameters does not provide a relevant benchmark.

- There is a methodological problem with the ablation experiment regarding conditional guidance. It is uncertain how the model learns to condition on $c$ when $r=0$. It would seem necessary to train a conditional diffusion model initially to generate accurate data points for minimizing the self-consistency loss.

- The results presented in Table 1 for CM and GD raise questions. It is confusing how these distillation techniques do not show a marked improvement over quick sampler methods that do not require training, such as DPM Solver and DPM Solver++.

**Questions:**

I would like to hear the authors' thoughts on the weaknesses identified above. In addition, I hope to get clarification on the derived DDIM with v-prediction in Figure 5. How is it different from DDIM?